# Effects of Consumption of Alcohol on Intraocular Pressure: Korea National Health and Nutrition Examination Survey 2010 to 2011

**DOI:** 10.3390/nu12082420

**Published:** 2020-08-12

**Authors:** Ji Eun Song, Joon Mo Kim, Mi Yeon Lee, Hye Joo Jang, Ki Ho Park

**Affiliations:** 1Department of Ophthalmology, Kangbuk Samsung Hospital, Sungkyunkwan University School of Medicine, Seoul 03181, Korea; jj.song@samsung.com (J.E.S.); j.heyjoojj@gmail.com (H.J.J.); 2Division of Biostatistics, Department of R&D Management, Kangbuk Samsung Hospital, Sungkyunkwan University School of Medicine, Seoul 03181, Korea; my7713.lee@samsung.com; 3Department of Ophthalmology, Seoul National University College of Medicine, Seoul 03080, Korea; kihopark@snu.ac.kr

**Keywords:** alcohol consumption, Koreans, open-angle glaucoma, IOP, sex

## Abstract

This study researched the association between alcohol consumption, intraocular pressure (IOP), and risk of open-angle glaucoma (OAG) using nationwide population-based cross-sectional data from the Korean population based survey. Information on alcohol intake was obtained by questionnaire and comprehensive ophthalmic examinations were performed. Among a total of 6057 participants, the prevalence of OAG was 4.4% (6.0% for men and 3.0% for women). Multivariate adjusted models showed that alcohol consumption showed significant relationship with changes in IOP. In sex-stratified analyses, alcohol consumption more than 2 times per week was associated with increased IOP in men without OAG, while in women with OAG drinking alcohol more than 4 times per week was associated with increased IOP. This study showed significant differences between men and women without glaucoma who consumed alcohol more than four times per week (*p*-value: 0.03). Our results suggest that alcohol consumption is associated with risk of elevated IOP depending on sex and presence of glaucoma in Koreans. Therefore, patients who need to control IOP should consider the effects of alcohol consumption.

## 1. Introduction

Glaucoma has been known as a chronic progressive optic neuropathy which can cause visual field loss resulting in irreversible blindness. The number of glaucoma patients over 40 years of age is estimated at 0.64 million people in the world, and is expected to rise to approximately 0.76 million by 2020 and 1.11 million by 2040 [1]. Although various glaucoma risk factors have been investigated, the most important factor is intraocular pressure (IOP). To date, the most effective treatment for glaucoma is reduction of IOP [2]. However, glaucoma may occur and progress even if the IOP decreases to the normal range, and glaucoma progression is not always associated with IOP itself. Recently, researchers have investigated extraocular risk factors that affect glaucoma development, such as body mass index (BMI) [3], metabolic syndrome [4], lipid level [5], and ocular factors like myopia [6,7], lens status [8], and lifestyle [9,10]. Alcohol produces physiologic effects that could be risk factors for glaucoma after consumption, including neurotoxicity [11,12], changes in vessel diameter [13,14] and fluctuations of blood osmotic pressure [10,15]. Although the mechanism mediating the relationship between alcohol consumption and IOP are not clearly understood [16,17], researchers have explored whether alcohol consumption is associated with the change of IOP. Some studies suggested that large amounts of alcohol intake increase IOP [18,19,20]. Kahn et al. [21] showed that more alcohol consumption was correlated with higher IOP and the deterioration of glaucomatous damage. On the contrary, other studies suggested that mild to moderate alcohol intake could reduce IOP [22,23]. The Beaver Dam Eye Study found no association between alcohol consumption and IOP change [24].

Although several studies have reported the relationship between alcohol drinking and IOP, most of these studies were performed on western people and response to alcohol varies among populations and individuals due to genetic and environmental factors. In South Korea, people tend to drink frequently after work in gatherings with colleagues, which is different from the drinking culture in Western countries. In this study, we investigate the association between alcohol intake and IOP with data from The Korea National Health and Nutrition Examination Survey (KNHANES), which is a population-based cross-sectional survey.

## 2. Materials and Methods

### 2.1. Data Source and Study Participants

This study was performed with data from the KNHANES 2010 to 2011, which is an ongoing, cross sectional, nationwide population-based survey. The data from the KNHANES are nationally representative of the civilian and non-institutionalized population in South Korea because KNHANES uses a stratified, multistage, probability-clustered sampling method according to groups based on age, sex, economic status, and geographic area, and uses a weighting scheme to conduct a detailed survey. Details related to KNHANES methods are reported elsewhere [25]. All studies using data from KNHANES were conducted according to the tenets of the Declaration of Helsinki, and all participants provided written informed consent. The KNHANES data are public and available online (http://knhanes.cdc.go.kr). All data from KNHANES are de-identified and the study protocol was approved by the Institutional Review Board of Kangbuk Samsung Hospital (kbsmc 2019-08-032).

A total of 12,356 participants aged 20 years or older were enrolled in the KNHANES 2010 to 2011. Of these, we excluded participants with any missing data and if they were aphakic or pseudophakic; if they had a history of refractive or retinal surgery, using anti-glaucoma agents, evidence of retinal detachment or macular degeneration, or abnormal liver function levels such as alanine aminotransferase (ALT) ≥ 300 or aspartate aminotransferase (AST) ≥ 300, to eliminate the effects of alcohol consumption on systemic changes. Participants with types of glaucoma other than open-angle glaucoma (OAG) or with any missing data were also excluded. A total of 6504 subjects were analyzed, including 288 OAG subjects.

### 2.2. Data Collection and Definitions of Variables

A general questionnaire was drew up to obtain subjects’ information about basic demographics, behaviors (smoking, alcohol consumption and physical activities), and medical conditions (history of physician diagnosed disease and current medications). All subjects were requested the information of medical history, alcohol intake, and smoking status. Based on average alcohol consumption per month for the year preceding the interview, subjects were categorized as drinkers (more than once a month) or non-drinkers. Also, subjects were categorized as current smokers (more than 100 cigarettes over their lifetime and current smoking status) or non-smokers. Impaired fasting glucose was defined as fasting blood glucose levels of 100 to 125 mg/dl. Diabetes mellitus (DM) was defined as a fasting glucose value greater than 126 mg/dl, use of insulin or oral hypoglycemic medications, or a history of DM. Prehypertension was defined as systolic blood pressure levels of 120 to 139 mmHg or diastolic blood pressure levels of 80 to 89 mmHg. Systemic hypertension was defined as systolic blood pressure greater than 140 mmHg, diastolic blood pressure greater than 90 mmHg, use of antihypertensive medication, or history of hypertension.

### 2.3. Ophthalmological Examination

All participants underwent ophthalmic examinations by certified ophthalmologists and participated in detailed ophthalmology-focused interviews. Slit lamp examination including evaluation of peripheral anterior chamber depth (PACD) with the van-Herick method (Haag-Streit model BQ-900; Haag-Streit AG, Koeniz, Switzerland). IOP measurement was performed with a Goldmann applanation tonometer in both eyes. However, because of the large number of samples, the data was analyzed with right eye IOP of all participants. Fundus photographs were obtained with a non-mydriatic digital retinal camera (TRC-NW6S; Topcon, Tokyo, Japan and Nikon D-80; Nikon, Tokyo, Japan) and visual field examination was conducted with frequency doubling technology (FDT; Humphrey Matrix; Carl Zeiss Meditec Inc., Dublin, CA, USA) with the N-30-1 screening test. The abnormal location of the test was defined if it was not assessed after two attempts at a contrast level by which 99% of the healthy population is identified. If two different test locations were abnormal, a visual field defect was detected in the eye.

Frequency doubling technology was carried out to subjects who met any of the following standard criteria for confirming suspected glaucoma: (1) IOP ≥ 22 mmHg; (2) horizontal or vertical cup-to-disc (C/D) ratio ≥ 0.5; (3) nonadherence to neuroretinal rim thickness in the following order: Inferior > Superior > Nasal > Temporal by quadrant, or the so-called ISNT rule; (4) presence of optic disc hemorrhage; or (5) presence of a retinal nerve fiber layer (RNFL) defect. Frequency doubling technology was re-checked if either the false-positive rate or the rate of fixation errors was greater than 30%, in which case the FDT was decided to be unreliable test for glaucoma classification.

### 2.4. Definitions of OAG and Control Groups

The diagnosis of OAG was based on the International Society of Geographical and Epidemiological Ophthalmology criteria and the results of previous studies [26,27,28]. Patients were decided as having OAG if they had an open anterior chamber angle with PACD > 1/4 corneal thickness based on the Van Herick method, and if they met any one of the following category I or II diagnostic criteria.

Category I criteria were applied to subjects with FDT perimetry results showing false-positive error and a fixation error of one or less. Criteria for diagnosis of glaucoma were (1) loss of neuroretinal rim with horizontal or vertical C/D ratio ≥ 0.7 or asymmetric C/D ratio ≥ 0.2 (both values determined by ≥ 97.5th percentile for the normal KNHANES population); (2) presence of optic disc hemorrhage; or (3) presence of an RNFL defect. In addition, the subjects had to have abnormal FDT perimetry results with at least one location of reduced sensitivity compatible with RNFL defect or optic disc appearance. Criteria II were applied to patients without FDT perimetry results, or with fixation errors or false-positive errors of two or more with (1) neuroretinal rim loss and vertical C/D ratio ≥ 0.9 or asymmetry of vertical C/D ratio ≥ 0.3 or (2) presence of an RNFL defect compatible with optic disc appearance.

The controls were participants who met all of the following criteria in their both eyes: (1) IOP ≤ 21 mmHg; (2) presence of an open angle (PACD > 1/4 corneal thickness); (3) non-glaucomatous optic disc (horizontal and vertical C/D ratio < 0.7 and inter-eye difference of horizontal and vertical C/D ratio < 0.2); (4) absence of RNFL defect or optic disc hemorrhage; and (5) optic disc not violating the ISNT rule.

After preliminary grading, more detailed grading was performed separately by another group of glaucoma specialists while blinded to other information about the participants. Cases of difference between preliminary and detailed grades were adjudicated by a third group of glaucoma specialists.

### 2.5. Statistical Analysis

Statistical analysis was conducted using STATA version 16.1 (StataCorp, College Station, TX, USA) to account for the complex sampling design. Strata and sampling units and weights were used to get standard errors (SEs) of mean and point estimates. All data analyses were carried out using weighted data, and SEs of the mean of population estimates were worked out using Taylor linearization methods. Participant characteristics were recorded for all of the sample using mean and SE for continuous variables and percentage, frequency, and SE for categorical variables.

Baseline characteristics of subjects and clinical parameters were compared between groups using Pearson’s chi-square test for categorical variables and general linear models (GLMs) for continuous variables. GLMs were used to evaluate relationships between alcohol consumption and IOP. To achieve more reliable results, values that were significantly different (*p* < 0.05) between the two groups at baseline were adjusted. For these models, we adjusted for sex, age, BMI, smoking, DM, systemic hypertension, and total cholesterol. After dividing the subjects into six groups according to frequency of alcohol consumption, we analyzed relationships between alcohol consumption and IOP. Logistic regression models were used to estimate odds ratios (OR) with 95% confidence intervals (CI). Subjects who consumed alcohol less than once per month were used as the reference. 95% CIs and β-coefficient values were obtained. And also ORs and 95% CIs for OAG risk were obtained. *p* values were two-tailed, and *p* < 0.05 was considered statistically significant except for multivariable analyses, for which Bonferroni adjustments were applied to correct type 1 errors despite multiple comparisons.

## 3. Results

### 3.1. Baseline Characteristics

A total of 6504 participants (6216 normal control, 288 OAG without treatment) were included in the analysis. Of this subset, 2983 (46%) were male and 180 (4.8%) had glaucoma, while 3521 (54%) were female and 108 (3.0%) had glaucoma. Table 1 shows demographic characteristics of participants. Glaucoma patients are more likely to have the following characteristics than non-glaucoma subjects: old age, high systolic/diastolic blood pressure, high serum glucose and low high-density lipoprotein (HDL), history of DM and systemic hypertension, and high IOP. And in both groups, subjects with high IOP (≥18 mmHg) were significantly more likely to have the following characteristics than subjects with low IOP (<18 mmHg): male sex, current smoker, drinker, high BMI, high waist circumference, high systolic/diastolic blood pressure, high total cholesterol level and low-density lipoprotein (LDL), and history of diabetes and systemic hypertension (Table 2). Subjects with high IOP (≥18 mmHg) are more likely to have the following characteristics in drinkers: high systolic/diastolic blood pressure, high serum glucose level, history of diabetes and systemic hypertension in male and high BMI in female (Table 3). Table 4 shows the prevalence of glaucoma according to alcohol consumption in males and females. No linear tendency was observed in the rate of glaucoma prevalence according to alcohol intake. The prevalence of glaucoma was lowest for men who drank alcohol two to four times a month and once a month for women, although the relationship was not statistically significant. Among participants who did not drink at all, men had a slightly higher glaucoma prevalence rate than women, with 6.33 (95% CI; 3.38–11.56) for men and 2.67 (95% CI; 1.51–4.65) for women.

### 3.2. The Association between Alcohol Consumption and IOP

Table 5 shows the β coefficient values that represent associations between alcohol intake and IOP. Male subjects without glaucoma who drank alcohol over 2 times per month had significant increases in IOP as the amount of alcohol consumed increased. For subjects who reported alcohol consumption 2–4 times per month (β coefficient values: 0.53 (95% CI, 0.08–0.98) in Models 1 and 2), 2–3 times per week (β coefficient values: 0.69 (95% CI, 0.27–1.11), 0.68 (95% CI, 0.26–1.10), 0.59 (95% CI, 0.18–1.01) in Models 1, 2 and 3, respectively), and over 4 times per week (β coefficient values: 0.82 (95% CI, 0.33–1.32), 0.82 (95% CI, 0.32–1.31), 0.73 (95% CI, 0.22–1.23) in Models 1, 2 and 3, respectively), IOP increases significantly as alcohol consumption increases. Women without glaucoma did not show significant changes in any adjusted models. Interestingly, Table 4 shows opposite results for men and women among glaucoma subjects. Male subjects with glaucoma who did not drink alcohol at all had increased IOP (β coefficient value: 1.84 (95% CI, 0.11–3.57) in Model 3), while female subjects who drank alcohol over 4 times per week had increased IOP (β coefficient value: 2.83 (95% CI, 0.28–5.38) in Model 3). Figure 1 represents mean IOP values according to alcohol consumption. Men with glaucoma who did not consume alcohol at all had the highest IOP at 16.21 (95% CI, 14.52–17.90) mmHg compared to that of drinkers, while female subjects who consumed alcohol more than four times per week had the highest IOP at of 16.96 (95% CI, 14.64–19.28) mmHg. On the other hand, among subjects without glaucoma who consumed alcohol over four times per week, male subjects had significantly higher IOP than women (*p*-value = 0.025).

## 4. Discussion

In the present study, we investigated the relationships between alcohol consumption and IOP in a South Korean population-based sample using KHANES data. Multivariate adjusted models showed that men without glaucoma who drank alcohol over 2 times per month exhibited significant increases in IOP according to amount of alcohol consumed and that men who drank alcohol over 2 times per week were significantly more likely to have high IOP (≥18 mmHg). In this study, we defined 18 mmHg as a high IOP value because it has been associated with the development of OAG [29,30]. In subjects with glaucoma, the IOPs of men who did not consume alcohol at all were higher than among the total sample, while we observed significant increases in IOP among women glaucoma subjects who drank alcohol over 4 times per week. Our results show that alcohol intake has various effects on IOP in Koreans, depending on whether they have glaucoma or not and are men or women.

Many researchers have investigated the effects of alcohol on the organs of the body, both by direct and indirect mechanisms [31]. Alcohol is eliminated primarily by oxidation in the liver, where it is degraded to acetaldehyde. When alcohol levels in the body are low, alcohol dehydrogenase (ADH) and aldehyde dehydrogenase (ALDH) are the principal enzymes utilized for metabolism, whereas when the alcohol level is high, the microsomal ethanol oxidizing system (MEOS) and catalase activate. During this process, the accumulation of toxic metabolites in the body causes cell damages that becomes greater as alcohol intake increases [32]. Because MEOS is induced by high and chronic alcohol exposure and is different from the other two metabolic systems, it could not only indirectly but directly aggravate the condition by impairing defense system [33]. These processes could make the environment of the optic nerve vulnerable. Neuroscientists and neuropathologists have reported that alcohol-induced liver injury is connected to brain damage by toxic and inflammatory mediators that are produced by the damaged liver that injure the brain and nervous system [34,35]. The metabolism of some elements, such as GABA (gamma aminobutyric acid) and NMDA (N-methyl-D-aspartate), can be affected by alcohol [36,37]. GABA, which functions as a neurotransmitter in retinal amacrine cells, ganglion cells, horizontal cells, and bipolar cells, influencing inner retina and optic nerve development, might be affected by ethanol [38]. NMDA, which could be modulated by ethanol, mediates the central nervous system, and therefore heavy alcohol drinking could have devastating effects like brain shrinkage [39]. Damage to optic nerve and retinal ganglion cells could be linked to risk of glaucoma. The intake of large amounts of alcohol causes lipid peroxidation in the body with vasodilation or vasoconstriction, and substances from these processes can accumulate in the vascular wall and act as risk factors for cardiovascular disease and systemic hypertension [40,41]. In turn, these effects on vessels and blood flow could affect the progression of glaucoma.

In this study, alcohol intake showed different effects on male subjects with and without glaucoma. Men with glaucoma who never consumed alcohol had higher IOP than men who drank alcohol, which was different from the pattern in subjects without glaucoma (Figure 1, Table 4). The cause may be that glaucomatous eyes are more sensitive to changes in osmotic pressure than normal eyes [16], perhaps due to functional damage to the aqueous outflow system characteristic of glaucomatous eyes, which are more sensitive to changes of net movement of water in the eye [42]. Therefore, any given increase in aqueous volume and osmotic pressure leads to a greater increase of IOP in glaucomatous eyes than normal eyes. Pexczon and Grant [43] reported that drinking as much as the equivalent of 50 mL of whiskey or 1 L of beer decreased IOP by 1 to 6 mmHg after 1 h in non-glaucoma subjects, while subjects with OAG showed a much greater decrease in IOP, as much as 30 mmHg (in one case). This effect of IOP reduction was started during the first 60 min and remained up to 180 min after alcohol consumption. It is not yet clear how alcohol affects IOP. One hypothesis posits that alcohol has osmotic effects similar to urea, mannitol, and glycerol [44,45]. Second, alcohol may suppress antidiuretic hormone, which decreases the net amount of water in the body including the aqueous humor [16]. Third, alcohol may directly affect the secretory cells of the ciliary processes, which could reduce the production of aqueous humor [17]. These mechanisms may temporarily affect the flow of aqueous humor. According to previous studies, it is likely that alcohol consumption causes IOP reduction in the short term. However the mechanism underlying the long-term effects of alcohol has not been identified clearly, and further research is needed.

The effects of alcohol intake on blood vessels varies, including vasodilation and vasoconstriction depending on the amount and duration of alcohol consumption. They also vary from individual to individual. Some studies have reported that light to moderate ethanol intake could reduce the risk of myocardial and cerebral infarction due to endothelium-dependent vasodilator responses of alcohol [46,47]. On the contrary, chronic high-dose ethanol could increase vasoconstriction [48]. Kojima et al. [49] showed that alcohol intake increases blood flow to the optic nerve head due to fast alcohol metabolism, especially in subjects with aldehyde dehydrogenase 2 gene mutations who exhibit slow alcohol metabolism. Due to vasodilation, light alcohol intake could reduce IOP. Such vasodilation may improve blood flow return and lower IOP by lowering episcleral venous pressure [50]. In spite of ethanol-induced vasodilator responses, the efficacy of alcohol as a therapy still remains uncertain. Seddon et al. [22] showed that IOP may be affected by drinking habits during longer periods as well, noting that males who never drank alcohol were 9.2 times more likely to have ocular hypertension than those who were daily alcohol drinkers. On the other hand, Bukiya et al. [51] reported that moderate to heavy alcohol intake reduces cerebral blood flow by decreasing cerebral artery diameter, along with vasoconstriction [52,53]. This dose-dependent effect of ethanol caused the diameter of cerebral arterioles to decrease, which could be explained not only by the physicochemical characteristic of osmolality but also cellular mechanisms such as inhibition of Ca^2+^ channel and voltage-gated K^+^ channels (BK channels) [54]. Such series of events may decrease blood flow to the brain, optic nerve, and ocular blood flow, which may be related to glaucoma [55]. However, the exact mechanism underlying the effects of alcohol on vessel function has not yet been identified, and further research is warranted.

Because males and females have different body composition [56], we stratified our data according to sex. We found that the association between alcohol consumption and IOP differed between sexes. The basic sex differences in body composition might influence the effects of alcohol on IOP. Frezza et al. [57] reported decreased gastric oxidation of ethanol, which indicates significantly decreased activity of ADH, and lower amounts of body water related to body fat in women who exhibited increased vulnerability to complications due to alcohol. Some neurologists reported that women are more vulnerable to effects of alcohol, especially on the brain [58]. Hypothetically, women have more parameters that tend to induce brain and neuroimmune system damage caused by alcohol [59]. Heavy alcohol drinking can cause nerve damage [60], and glaucoma is a neuropathy of the optic nerve. Other studies [61] have revealed differences in risk factors and prevalence of glaucoma between men and women. However, the exact mechanism of sex-specific effects remains unclear, and it would be interesting to analyze the effects of alcohol according to sex in association with glaucoma progression in further research. To elucidate this topic, prospectively designed large scale studies are needed.

There were some limitations in our study. First, alcohol consumption was assessed only once based on a self-administered questionnaire and classified by frequency rather than amount or types of alcohol due to lack of data. This could lead to misclassifications, although previous epidemiologic studies have shown that self-reports of alcohol consumption are reliable [62]. Additionally, because we used data from a one-year survey of alcohol consumption recording the period immediately before the subjects participated in our study, their lifetime drinking habits and the impact of past alcohol drinking were unknown. In addition, to date, few studies have analyzed the effects of drinking on systemic disease according to different kinds of alcohol such as wine, beer, vodka, and rum [63,64]. It is believed that different types of alcohol may have different effects on IOP and glaucoma as well as systemic diseases. The more detailed the analysis, the more accurate the study would be, but it is difficult to include so many variables. Second, this study was a cross-sectional survey and used data from KNHANES. Therefore, our results may be not generalizable across ethnicities and we were unable to evaluate longitudinal changes and progression of glaucoma based on dose-related alcohol consumption. Therefore, further longitudinal studies in different geographic regions are required. The response to alcohol might vary from person to person, but we were unable to analyze this variable. Genetic variation, such as ALDH2 (aldehyde dehydrogenase 2) mutations (the so-called ‘flushing gene’) is highly relevant to alcohol metabolism [32,65]. Some studies have shown that the risk of systemic hypertension is up to twice as likely to occur in people with inactive or mutant ALDH2 alleles, which leads to variation in reactions to and unfavorable effects of alcohol [66]. Genetic variation is thought to be associated not only with systemic disease but also with risk of glaucoma. Further research and investigation are needed to figure out the relationships between type of alcohol, genetic variation, and risk of glaucoma or other ophthalmic diseases. Finally, we evaluated the status of anterior chamber angle using Van-Herick methods rather than gonioscopy. Despite these limitations, our study included a large sample size and had a high participation rate, and was representative of the whole population in South Korea. Our research also includes ophthalmic examinations such as Goldmann applanation tonometer and slit lamp examination that were performed by trained ophthalmologists.

One of the most important factors causing glaucomatous damage is higher IOP than the optic nerve can withstand. The threshold of response to each factor depends on sex, age, ethnicity, genetic variation, and other elements. In this population-based study using data from KHNANES, we identified an association between alcohol consumption and changes in IOP. Daily alcohol intake was significantly associated with IOP elevation in patients with glaucoma, especially women. On the other hand, mild consumption of alcohol (once to four times per month) could be helpful to decrease IOP, especially among men, regardless of glaucoma. Although we did not observe significant results regarding the prevalence of glaucoma according to alcohol intake, further research is needed to determine whether changes in IOP caused by alcohol intake are related to the progression of glaucoma. Alcohol intake shows significant effects on IOP, but the effect was different according to alcohol dose and sex. Further longitudinal studies are required to clearly identify mechanisms and determine whether sex differences underlie the effects of alcohol effects on IOP.

## Figures and Tables

**Figure 1 nutrients-12-02420-f001:**
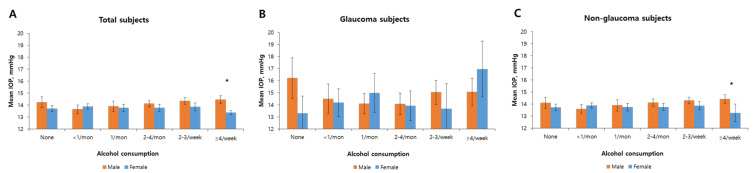
Difference in intraocular pressure (IOP) according to alcohol consumption in total subjects (*n* = 6504) (**A**), subjects with glaucoma (*n* = 288) (**B**), and without glaucoma (*n* = 6216) (**C**). Data are presented as mean values (95% confidence intervals). The mean IOPs of men in total (mean IOP (95% CI) (IOP); 14.45 (14.11–14.79), *p*-value; 0.048) and non-glaucoma group (mean IOP (95% CI); 14.3 (13.98–14.62), *p*-value; 0.03) who consumed alcohol over four times per week are significantly higher than those of women (mean IOP in total subjects (95% CI); 13.37 (12.65–14.09), mean IOP in non-glaucoma subjects (95% CI); 13.27 (12.55–14)). Asterisks indicate significant differences at *p* < 0.05.

**Table 1 nutrients-12-02420-t001:** Baseline characteristics of the study participants.

Variables	Total (*n* = 6504)		*p* Value	Male (*n* = 2983, 45.9%)		*p* Value	Female (*n* = 3521, 54.1%)		*p* Value
	Non-Glaucoma (*n* = 6216, 96.1%)	Glaucoma (*n* = 288, 3.9%)		Non-Glaucoma (*n* = 2803, 95.2%)	Glaucoma (*n* = 180; 4.8%)		Non-Glaucoma (*n* = 3413, 97.0%)	Glaucoma (*n* = 108, 3.0%)	
Age, years	41.1 (0.3)	49.2 (1.2)	<0.001	41 (0.4)	49.2 (1.4)	<0.001	41.3 (0.3)	49.1 (1.9)	<0.001
Current smoker, %	30 (0.8)	31.5 (3.44)	0.678	50.5 (1.21)	44.8 (4.45)	0.227	7.6 (0.58)	7.4 (3.08)	0.958
Drinker, %	68 (0.78)	68.8 (3.35)	0.814	83.2 (0.87)	78.9 (3.9)	0.248	51.5 (1.09)	50.7 (6.2)	0.907
BMI, kg/m^2^	23.7 (0.1)	23.9 (0.2)	0.496	24.2 (0.1)	23.7 (0.2)	0.032	23.1 (0.1)	24.2 (0.5)	0.025
Waist circumference, cm	81 (0.2)	82.2 (0.7)	0.095	84.5 (0.2)	83.5 (0.7)	0.201	77.1 (0.2)	79.9 (1.3)	0.044
Systolic blood pressure, mmHg	116.3 (0.3)	122.8 (1.3)	<0.001	119.5 (0.3)	123.9 (1.6)	0.008	112.7 (0.4)	120.9 (2)	<0.001
Diastolic blood pressure, mmHg	76.7 (0.2)	80.1 (0.8)	<0.001	79.7 (0.3)	81.7 (1.1)	0.078	73.3 (0.2)	77.2 (1.1)	0.001
Serum glucose, mg/dL	94.9 (0.3)	99.9 (1.9)	0.009	97.5 (0.5)	102 (2.8)	0.111	92.1 (0.3)	96.2 (1.7)	0.02
Total cholesterol, mg/dL	187.1 (0.6)	188.6 (3.4)	0.666	188.8 (0.9)	186.4 (4.7)	0.621	185.3 (0.7)	192.6 (3.7)	0.049
HDL-C, mg/dL	53.3 (0.2)	51 (1.02)	0.026	50 (0.3)	48.8 (1.3)	0.389	56.9 (0.3)	54.8 (1.3)	0.121
LDL-C, mg/dL	111.9 (0.8)	110.1 (4.5)	0.694	113.9 (1.1)	109 (5.5)	0.387	109.1 (1.2)	113.1 (7.9)	0.605
Triglycerides, mg/dL	132.2 (1.9)	154 (14.4)	0.132	158.5 (3.2)	172 (21.7)	0.536	103.4 (1.6)	121.5 (7.2)	0.015
Diabetic, %	21.8 (0.74)	31.1 (3.24)	0.002	27.1 (1.11)	36.5 (4.74)	0.038	16.1 (0.8)	21.7 (4.14)	0.147
Hypertension, %	42.1 (0.92)	58.8 (3.74)	<0.001	52.9 (1.29)	62.3 (4.86)	0.064	30.2 (1.02)	52.4 (5.72)	<0.001
IOP (mmHg)	14 (0.1)	14.5 (0.2)	0.019	14.1 (0.1)	14.7 (0.2)	0.021	13.8 (0.1)	14.1 (0.3)	0.391

Data are presented as mean (SE) for continuous variables and as percentage (SE) for categorical variables. BMI, body mass index; CI, confidence interval; DM, diabetes mellitus; HDL-C, high-density lipoprotein cholesterol; IOP, intraocular pressure; LDL-C, low-density lipoprotein cholesterol; SE, standard error.

**Table 2 nutrients-12-02420-t002:** Participant characteristics according to intraocular pressure (IOP) between non-glaucoma and glaucoma group.

Variables	Total (*n* = 6504)		*p* Value	Non-Glaucoma (*n* = 6216, 96.1%)		*p* Value	Glaucoma (*n* = 288, 3.9%)		*p* Value
	IOP ≥ 18 mmHg (*n* = 730, 11.3%)	IOP < 18 mmHg (*n* = 5774, 88.7%)		IOP ≥ 18 mmHg (*n* = 683; 11.1%)	IOP < 18 mmHg (*n* = 5533, 88.9%)		IOP ≥ 18 mmHg (*n* = 47, 16.9%)	IOP < 18 mmHg (*n* = 241, 83.1%)	
Age, years	41.8 (0.6)	41.4 (0.3)	0.523	41.2 (0.6)	41.1 (0.3)	0.905	51.8 (2.3)	48.6 (1.3)	0.236
Male, %	59.1 (2.2)	51.9 (0.75)	0.004	58.6 (2.3)	51.5 (0.76)	0.056	67.6 (8.46)	63.5 (3.64)	0.665
Current smoker, %	35.1 (2.43)	29.4 (0.82)	0.021	34.6 (2.56)	29.5 (0.82)	0.044	43.3 (9.12)	29 (3.7)	0.13
Drinker, %	73.7 (2.05)	67.3 (0.8)	0.005	73.3 (2.15)	67.4 (0.81)	0.011	79.6 (7.23)	66.7 (3.55)	0.14
BMI, kg/m^2^	24.3 (0.2)	23.6 (0.1)	0.002	24.3 (0.2)	23.6 (0.1)	0.002	24.1 (0.6)	23.8 (0.3)	0.592
Waist circumference, cm	82.7 (0.5)	80.8 (0.2)	0.001	82.6 (0.5)	80.8 (0.2)	0.001	84.6 (2.3)	81.8 (0.7)	0.226
Systolic blood pressure, mmHg	120 (0.7)	116.1 (0.3)	<0.001	119.6 (0.7)	115.9 (0.3)	<0.001	126.9 (3.6)	121.9 (1.3)	0.181
Diastolic blood pressure, mmHg	78.9 (0.5)	76.5 (0.2)	<0.001	78.7 (0.5)	76.4 (0.2)	<0.001	82.5 (2.5)	79.6 (0.8)	0.267
Serum glucose, mg/dL	97.3 (1.2)	94.8 (0.3)	0.052	97.4 (1.3)	94.6 (0.3)	0.033	95.7 (3.7)	100.8 (2.2)	0.249
Total cholesterol, mg/dL	191.4 (1.6)	186.6 (0.6)	0.004	191.4 (1.7)	186.6 (0.6)	0.007	192.5 (7.2)	187.8 (3.7)	0.552
HDL-C, mg/dL	52.4 (0.6)	53.3 (0.2)	0.138	52.4 (0.6)	53.4 (0.2)	0.1	52.5 (2.1)	50.7 (1.1)	0.417
LDL-C, mg/dL	120.1 (2.2)	110.8 (0.8)	<0.001	120.8 (2.3)	110.9 (0.8)	<0.001	111.7 (6.7)	109.7 (5.3)	0.806
Triglycerides, mg/dL	143.4 (5.4)	131.7 (2.1)	0.049	142.8 (5.7)	130.8 (2)	0.056	152.5 (18.2)	154.2 (17)	0.945
Diabetic, %	25.9 (2.05)	21.7 (0.76)	0.038	25.7 (2.18)	21.3 (0.77)	0.04	29.3 (7.7)	31.5 (3.71)	0.811
Hypertension, %	53.1 (2.14)	41.4 (0.95)	<0.001	52.1 (2.21)	40.8 (0.98)	<0.001	70.2 (8.38)	56.5 (3.94)	0.151
IOP (mmHg)	18.7 (0.04)	13.4 (0.1)	<0.001	18.7 (0.04)	13.4 (0.1)	<0.001	18.6 (0.2)	13.6 (0.2)	<0.001

Data are presented as mean (SE) for continuous variables and as percentage (SE) for categorical variables. BMI, body mass index; CI, confidence interval; DM, diabetes mellitus; HDL-C, high-density lipoprotein cholesterol; IOP, intraocular pressure; LDL-C, low-density lipoprotein cholesterol; SE, standard error.

**Table 3 nutrients-12-02420-t003:** Participant characteristics according to intraocular pressure (IOP) in drinkers.

Variables	Drinker		*p* Value
	IOP ≥ 18 mmHg	IOP < 18 mmHg	
Male			
Number, %	328 (13.32)	2113 (86.7)	
Age, years	41.73 (0.82)	40.69 (0.41)	0.227
Current smoker, %	170 (56.39)	996 (52.11)	0.255
BMI, kg/m^2^	24.5 (0.28)	24.21 (0.08)	0.301
Waist circumference, cm	85.1 (0.75)	84.51 (0.26)	0.450
Systolic blood pressure, mmHg	123.76 (0.93)	119.67 (0.38)	<0.001
Diastolic blood pressure, mmHg	82.13 (0.66)	79.87 (0.32)	0.002
Serum glucose, mg/dL	100.64 (1.98)	97.35 (0.53)	0.112
Total cholesterol, mg/dL	193.56 (2.42)	187.92 (1.02)	0.023
HDL-C, mg/dL	50.69 (0.71)	50.55 (0.35)	0.849
LDL-C, mg/dL	121.59 (3.26)	110.93 (1.23)	0.002
Triglycerides, mg/dL	169.13 (9.66)	165.57 (4.13)	0.734
Diabetic, %	121 (33.04)	650 (26.81)	0.043
Hypertension, %	213 (62.33)	1215 (53.44)	0.011
IOP (mmHg)	18.83 (0.06)	13.52 (0.08)	<0.001
Female			
Number, %	185 (10.3)	1518 (89.7)	
Age, years	39.74 (1.08)	39.94 (0.38)	0.866
Current smoker, %	17 (10.51)	132 (10.58)	0.981
BMI, kg/m^2^	23.83 (0.39)	22.97 (0.11)	0.031
Waist circumference, cm	78.64 (1.1)	76.74 (0.31)	0.097
Systolic blood pressure, mmHg	113.52 (1.22)	112.51 (0.56)	0.439
Diastolic blood pressure, mmHg	74.53 (0.81)	73.23 (0.36)	0.130
Serum glucose, mg/dL	93.4 (1.24)	92.07 (0.44)	0.319
Total cholesterol, mg/dL	183.12 (2.47)	183.72 (1.12)	0.821
HDL-C, mg/dL	56.95 (1.33)	58.37 (0.39)	0.308
LDL-C, mg/dL	108.12 (4.31)	105.86 (1.79)	0.618
Triglycerides, mg/dL	100.31 (4.64)	101.16 (2.19)	0.868
Diabetic, %	35 (16.35)	274 (16.08)	0.94
Hypertension, %	69 (33.56)	480 (28.13)	0.165
IOP (mmHg)	18.6 (0.08)	13.21 (0.08)	<0.001

Data are presented as mean (SE) for continuous variables and as percentage (SE) for categorical variables. BMI, body mass index; CI, confidence interval; DM, diabetes mellitus; HDL-C, high-density lipoprotein cholesterol; IOP, intraocular pressure; LDL-C, low-density lipoprotein cholesterol; SE, standard error.

**Table 4 nutrients-12-02420-t004:** The prevalence of glaucoma according to alcohol consumption and sex.

Alcohol Drinking	Total	Male	Female
None	3.78 (2.49–5.69)	6.33 (3.38–11.56)	2.67 (1.51–4.65)
<1 time/month	3.86 (2.72–5.45)	5.65 (3.37–9.32)	3.21 (2.07–4.95)
1 time/month	4.06 (2.58–6.33)	6.68 (3.71–11.73)	2.36 (1.25–4.41)
2–4 times/month	3.31 (2.46–4.45)	3.51 (2.39–5.12)	3.01 (1.88–4.79)
2–3 times/week	4.78 (3.57–6.39)	5.09 (3.58–7.18)	3.82 (2.18–6.61)
≥4 times/week	4.42 (2.78–6.94)	4.71 (2.87–7.65)	2.56 (0.89–7.17)
Total	3.93 (3.35–4.61)	4.78 (3.94–5.8)	2.98 (2.36–3.74)

Data are presented as percentage (95% CI) for categorical variables. CI, confidence interval.

**Table 5 nutrients-12-02420-t005:** Association between alcohol consumption and intraocular pressure (IOP).

Alcohol Drinking	Unadjusted β Coefficient			Model 1 *			Model 2 ^†^			Model 3 ^‡^		
	Total β (95% CI)	Male β (95% CI)	Female β (95% CI)	Total β (95% CI)	Male β (95% CI)	Female β (95% CI)	Total β (95% CI)	Male β (95% CI)	Female β (95% CI)	Total β (95% CI)	Male β (95% CI)	Female β (95% CI)
IOP (β coefficient); Non-glaucoma subjects												
None	0.03 (−0.24 to 0.3)	0.51 (−0.07 to 1.08)	−0.15 (−0.46 to 0.15)	0.01 (−0.27 to 0.28)	0.51 (−0.06 to 1.09)	−0.17 (−0.47 to 0.14)	0.01 (−0.27 to 0.28)	0.52 (−0.06 to 1.09)	−0.17 (−0.47 to 0.14)	0.001 (−0.27 to 0.27)	0.48 (−0.12 to 1.07)	−0.16 (−0.47 to 0.14)
<1 time/month	0 (reference)	0 (reference)	0 (reference)	0 (reference)	0 (reference)	0 (reference)	0 (reference)	0 (reference)	0 (reference)	0 (reference)	0 (reference)	0 (reference)
1 time/month	−0.002 (−0.31 to 0.31)	0.3 (−0.27 to 0.87)	−0.14 (−0.47 to 0.19)	−0.02 (−0.33 to 0.29)	0.33 (−0.24 to 0.91)	−0.13 (−0.47 to 0.2)	−0.02 (−0.33 to 0.29)	0.33 (−0.24 to 0.91)	−0.14 (−0.48 to 0.19)	−0.06 (−0.37 to 0.25)	0.3 (−0.28 to 0.88)	−0.19 (−0.52 to 0.14)
2–4 times/month	0.19 (−0.07 to 0.44)	0.54 (0.08 to 0.99)	−0.12 (−0.45 to 0.21)	0.1 (−0.16 to 0.36)	0.53 (0.08 to 0.98)	−0.09 (−0.43 to 0.24)	0.1 (−0.16 to 0.36)	0.53 (0.08 to 0.98)	−0.09 (−0.43 to 0.25)	0.06 (−0.2 to 0.32)	0.46 (−0.00 to 0.91)	−0.1 (−0.44 to 0.24)
2–3 times/week	0.4 (0.12 to 0.67)	0.71 (0.29 to 1.13)	−0.02 (−0.43 to 0.39)	0.24 (−0.05 to 0.53)	0.69 (0.27 to 1.11)	−0.01 (−0.42 to 0.4)	0.24 (−0.05 to 0.54)	0.68 (0.26 to 1.10)	0.0 (−0.42 to 0.43)	0.19 (−0.11 to 0.48)	0.59 (0.18 to 1.01)	−0.02 (−0.45 to 0.41)
≥4 times/week	0.45 (0.1 to 0.81)	0.82 (0.33 to 1.32)	−0.6 (−1.34 to 0.14)	0.28 (−0.1 to 0.66)	0.82 (0.33 to 1.32)	−0.66 (−1.43 to 0.1)	0.28 (−0.1 to 0.66)	0.82 (0.32 to 1.31)	−0.64 (−1.41 to 0.13)	0.25 (−0.14 to 0.63)	0.73 (0.22 to 1.23)	−0.53 (−1.28 to 0.21)
*p* for trend	0.001	0.002	0.611	0.049	0.002	0.698	0.052	0.003	0.764	0.116	0.011	0.741
IOP (β coefficient); Glaucoma subjects												
None	0.47 (−1.19 to 2.14)	1.71 (−0.4 to 3.82)	−0.89 (−2.75 to 0.98)	0.43 (−1.09 to 1.95)	1.79 (−0.01 to 3.59)	−0.98 (−2.68 to 0.73)	0.43 (−1.1 to 1.95)	1.8 (−0.01 to 3.58)	−0.98 (−2.69 to 0.73)	0.49 (−1 to 1.98)	1.84 (0.11 to 3.57)	−0.82 (−2.53 to 0.9)
<1 time/month	0 (reference)	0 (reference)	0 (reference)	0 (reference)	0 (reference)	0 (reference)	0 (reference)	0 (reference)	0 (reference)	0 (reference)	0 (reference)	0 (reference)
1 time/month	0.1 (−1.2 to 1.39)	−0.42 (−1.93 to 1.1)	0.8 (−1.19 to 2.8)	0.01 (−1.36 to 1.39)	−0.33 (−1.76 to 1.1)	0.91 (−1.07 to 2.89)	0.03 (−1.35 to 1.41)	−0.37 (−1.81 to 1.07)	0.91 (−1.07 to 2.89)	0.18 (−1.15 to 1.51)	0.21 (−1.52 to 1.1)	1.08 (−0.95 to 3.11)
2–4 times/month	−0.29 (−1.38 to 0.8)	−0.43 (−1.95 to 1.09)	−0.27 (−1.98 to 1.44)	−0.51 (−1.61 to 0.59)	−0.65 (−2.05 to 0.75)	−0.28 (−1.86 to 1.31)	−0.51 (−1.62 to 0.59)	−0.63 (−2.07 to 0.82)	−0.28 (−1.83 to 1.27)	−0.5 (−1.63 to 0.63)	−0.6 (−2.06 to 0.86)	−0.17 (−1.7 to 1.36)
2–3 times/week	0.48 (−0.88 to 1.84)	0.54 (−1.16 to 2.24)	−0.5 (−2.89 to 1.89)	0.21 (−1.19 to 1.61)	0.57 (−0.92 to 2.06)	−0.14 (−2.44 to 2.17)	0.25 (−1.11 to 1.61)	0.49 (−0.95 to 1.93)	−0.16 (−2.57 to 2.24)	−0.01 (−1.34 to 1.32)	0.21 (−1.22 to 1.65)	−0.07 (−2.45 to 2.31)
≥4 times/week	0.91 (−0.53 to 2.35)	0.56 (−1.12 to 2.24)	2.79 (0.19 to 5.38)	0.5 (−0.9 to 2.03)	0.64 (−1.01 to 2.28)	2.55 (−0.0 to 5.1)	0.6 (−0.85 to 2.06)	0.56 (−1.07 to 2.2)	2.55 (−0.0 to 5.1)	0.42 (−0.99 to 1.82)	0.37 (−1.24 to 1.98)	2.83 (0.28 to 5.38)
*p* for trend	0.546	0.837	0.579	0.982	0.823	0.378	0.967	0.596	0.336	0.627	0.331	0.305

Multivariate logistic regression. ***** Model 1: adjusted for age, sex and BMI; **^†^** Model 2: adjusted for age, sex, BMI and smoking; **^‡^** Model 3: adjusted for age, sex, BMI, smoking, DM, systemic hypertension and total cholesterol; BMI: body mass index, CI: confidence interval, DM: diabetes mellitus.

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
