# Peer review of "Effects of Consumption of Alcohol on Intraocular Pressure: Korea National Health and Nutrition Examination Survey 2010 to 2011"

_nutrients, 2020, doi:10.3390/nu12082420_

Round 1
Reviewer 1 Report
1.Explain in brief what is a biological mechanism of consumption of alcohol with Intraocular pressure?
2. Which significant biomarkers are associate for driving to development of intraocular pressure (IOP) association with alcohol consumption?
3. Add Cyp1b1 expression level changes for total subjects, alcohol consumption group, non- alcohol consumption group between male against female population as Cyp1b1 is one major enzyme which leads to glaucoma.
4. Add body mass index level changes for total subjects, alcohol consumption group, non- alcohol consumption group between male against female population.
Author Response
We are pleased to have the chance to revise our manuscript for publication in Nutrients. We appreciate the kind and constructive comments from the reviewers. We have provided point-by-point answers to their specific comments and indicated the revisions in the manuscript file in highlight. We also underlined the changes of some typos we found. Please let us know if we need any additional corrections. We hope that our revised manuscript is now suitable for publication.

Reviewer 2 Report
This is a well conducted study and well written paper, with actionable findings. I have two suggestions.
- Please clarify in the methods section how IOP was handled in the analyses (mean of both eyes, use of highest IOP or only data for only one eye, use of data for both eyes with adjustment for correlation between the two eyes, etc).
- I suggest deleting Table 5 as the data do not add substantially to the results or conclusions.
Author Response
We are pleased to have the chance to revise our manuscript for publication in Nutrients. We appreciate the kind and constructive comments from the reviewers. We have provided point-by-point answers to their specific comments and indicated the revisions in the manuscript file in highlight. We also underlined the changes of some typos we found. Please let us know if we need any additional corrections. We hope that our revised manuscript is now suitable for publication.

This manuscript is a resubmission of an earlier submission. The following is a list of the peer review reports and author responses from that submission.